# Modification of Mesenchymal Stem/Stromal Cell-Derived Small Extracellular Vesicles by Calcitonin Gene Related Peptide (CGRP) Antagonist: Potential Implications for Inflammation and Pain Reversal

**DOI:** 10.3390/cells13060484

**Published:** 2024-03-10

**Authors:** Kevin Liebmann, Mario A. Castillo, Stanislava Jergova, Thomas M. Best, Jacqueline Sagen, Dimitrios Kouroupis

**Affiliations:** 1Department of Orthopedics, UHealth Sports Medicine Institute, Miller School of Medicine, University of Miami, Miami, FL 33146, USA; kxl403@med.miami.edu (K.L.); mxc2845@med.miami.edu (M.A.C.); txb440@med.miami.edu (T.M.B.); 2Diabetes Research Institute & Cell Transplant Center, Miller School of Medicine, University of Miami, Miami, FL 33136, USA; 3Miami Project to Cure Paralysis, Miller School of Medicine, University of Miami, Miami, FL 33136, USA; sjergova@med.miami.edu (S.J.); jsagen@med.miami.edu (J.S.)

**Keywords:** mesenchymal stem/stromal cells (MSC), synovium, infrapatellar fat pad (IFP), CD10 (neprilysin), calcitonin gene-related peptide (CGRP), small extracellular vesicles (sEVs), immunomodulation, analgesia, osteoarthritis (OA)

## Abstract

During the progression of knee osteoarthritis (OA), the synovium and infrapatellar fat pad (IFP) can serve as source for Substance P (SP) and calcitonin gene-related peptide (CGRP), two important pain-transmitting, immune, and inflammation modulating neuropeptides. Our previous studies showed that infrapatellar fat pad-derived mesenchymal stem/stromal cells (MSC) acquire a potent immunomodulatory phenotype and actively degrade Substance P via CD10 both in vitro and in vivo. On this basis, our hypothesis is that CD10-bound IFP-MSC sEVs can be engineered to target CGRP while retaining their anti-inflammatory phenotype. Herein, human IFP-MSC cultures were transduced with an adeno-associated virus (AAV) vector carrying a GFP-labelled gene for a CGRP antagonist peptide (aCGRP). The GFP positive aCGRP IFP-MSC were isolated and their sEVs’ miRNA and protein cargos were assessed using multiplex methods. Our results showed that purified aCGRP IFP-MSC cultures yielded sEVs with cargo of 147 distinct MSC-related miRNAs. Reactome analysis of miRNAs detected in these sEVs revealed strong involvement in the regulation of target genes involved in pathways that control pain, inflammation and cartilage homeostasis. Protein array of the sEVs cargo demonstrated high presence of key immunomodulatory and reparative proteins. Stimulated macrophages exposed to aCGRP IFP-MSC sEVs demonstrated a switch towards an alternate M2 status. Also, stimulated cortical neurons exposed to aCGRP IFP-MSC sEVs modulate their molecular pain signaling profile. Collectively, our data suggest that yielded sEVs can putatively target CGRP in vivo, while containing potent anti-inflammatory and analgesic cargo, suggesting the promise for novel sEVs-based therapeutic approaches to diseases such as OA.

## 1. Introduction

Accumulating evidence implicates a potential role for calcitonin gene-related peptide (CGRP) in both the central and peripheral mechanisms of OA-related pain [1,2,3,4,5]. CGRP nerve fiber density is increased in a time-dependent fashion following increased nerve growth factor (NGF) levels and is tissue specific to the OA joint itself [6]. Higher levels of CGRP can then act on a wide variety of cell types in the joint. It can inhibit osteoclasto genesis, thus contributing to bone sclerosis and disease progression [7]. CGRP released by nerve fibers can also act on the fibers themselves, enhancing pain sensitization. CGRP release is calcium dependent, so overexpression of calcium channel modulators can reduce the pain threshold in rat MIA models of arthritis via downregulation of CGRP pathways [8].

The analgesic effect of CGRP_8–37_, a competitive antagonist of CGRP, has been shown in several pain models [9,10,11,12,13,14,15,16,17,18]. On this basis, Sagen et al. have recently generated and sequenced a DNA fragment encoding CGRP_8–37_ cDNA along with viral vectors (adeno-associated virus; AAV and lentivirus) that can be utilized in the development of gene therapeutic approaches to reduce chronic pain (Figure 1). AAV is widely considered the top clinically acceptable candidate due to low toxicity and the ability to provide continued transgene expression and long-term production of desired therapeutic molecules. There have now been over 130 clinical gene therapy trials involving AAVs, with an overall excellent safety profile [19].

In addition to the established roles of Substance P (SP) in nociception [20], recent evidence supports its participation in the modulation of local neurogenic inflammatory/immune responses [21,22]. SP increases vascular permeability favoring immune cell infiltration, while directly affecting macrophage phenotypic polarization and migration to sites of inflammation [23,24,25]. Meanwhile, regulation of SP activity is performed partly by cell membrane-bound neutral endopeptidase CD10 (neprilysin) [26], which is expressed by multiple MSC types [27]. Our previous work demonstrated that human IFP-MSC acquire a potent immunomodulatory phenotype and actively degrade SP via CD10 in vitro and in vivo [28,29]. Importantly, our recent data indicated that CD10-bound small extracellular vesicles (sEVs) possess immunomodulatory miRNA attributes with strong immunomodulatory and anabolic effects favoring joint homeostasis in vivo [30]. sEVs are nanometer-sized membranous particles that are immunologically inert and contain many constituents of a cell, including DNA, RNA, lipids, metabolites, and cytosolic and cell-surface proteins. MSC-derived sEVs mediate various biological functions attributed to MSC, such as tissue regeneration, intercellular communication, modulation of immunity and cell signaling [31]. On this basis, sEVs can be manipulated and engineered to deliver robust therapeutic molecules targeting inflammation and pain reversal in vivo.

In this study, we seek to engineer the sEVs of CD10-bound aCGRP IFP-MSC so they could also target CGRP in addition to Substance P. We transduced MSC with a CGRP_8–37_ gene construct for the purpose of developing and evaluating innovative sEVs-based strategies for inflammation and pain reversal.

## 2. Materials and Methods

### 2.1. Isolation, Culture and Expansion of IFP-MSC

All experiments using human cells were performed in accordance with the relevant guidelines and regulations. Human IFP-MSC were isolated from IFP tissue obtained from de-identified, non-arthritic patients [n = 5; two males 26 and 48 years old (labelled as MSC1 and MSC2), and three females 22, 42, 44 years old (labelled as MSC3, MSC4, MSC5)] undergoing elective knee arthroscopy at the Lennar Foundation Medical Center at the University of Miami. All procedures were carried out following approval by the University of Miami Institutional Review Board not as human research (based on the nature of the samples as discarded tissue). IFP tissue (5–10 cc) was mechanically dissected and washed repeatedly with Dulbecco’s Phosphate Buffered Saline (DPBS; Sigma-Aldrich, St Louis, MO, USA), followed by enzymatic digestion using 235 U/mL Collagenase I (Worthington Industries, Columbus, OH, USA) diluted in DPBS and 1% bovine serum albumin (Sigma) for 2 h at 37 °C with agitation. Enzymatic digestion was inactivated with complete media with DMEM low glucose (1 g/L) GlutaMAX (ThermoFisher Scientific, Waltham, MA, USA) containing 10% fetal bovine serum (FBS; VWR, Radnor, PA, USA), washed and seeded at a density of 1 × 10^6^ cells/175 cm^2^ flask in chemically reinforced (Ch-R) medium. Complete Ch-R medium was prepared by mixing Mesenchymal Stem Cell Growth Medium 2 with supplement provided according to manufacturer’s instructions (PromoCell, Heidelberg, Germany). At 48 h post-seeding, non-adherent cells were removed by DPBS rinsing and fresh media were replenished accordingly. MSC were cultured at 37 °C, 5% (*v*/*v*) CO_2_ until 80% confluent as passage 0 (P0), then passaged at a 1:5 ratio until P3, detaching them with TrypLE™ Select Enzyme 1X (Gibco, ThermoFisher Scientific) and assessing cell viability with 0.4% (*w*/*v*) Trypan Blue (Invitrogen, ThermoFisher Scientific).

### 2.2. Generation of AAV Vector Containing GFP-Labeled CGRP Antagonist Gene

CGRP_8–37_ is a truncated polypeptide of CGRP that serves as a competitive antagonist of CGRP (aCGRP). To create a gene construct, human CGRP cDNA was purchased from (Open Biosystems, ThermoFisher Scientific). The PCR fragment was ligated to pGEMT-Easy cloning vector and CGRP_8–37_ fragment subcloned downstream of peptidylglycine-amidating monooxygenase (ssPAM/pGEMT) signal sequence to allow CGRP_8–37_ to be amidated and secreted. ClaI-BamHI fragment was subcloned into pAAV-EGFP-WPRE viral vector. The ssPAM-CGRP_8–37_ was subcloned to AAV-EGFP-WPRE and lenti-EGFP-WPRE vectors and viral particles were developed by Miami Project Viral Vector Core (University of Miami).

### 2.3. IFP-MSC Transduction and Cell Sorting for GFP Positive aCGRP IFP-MSC

AAV- aCGRP-GFP was used to transduct 50 × 10^6^ P1 IFP-MSC (n = 5). Transducted cultures were maintained at 37 °C, 5% (*v*/*v*) CO_2_ for 7 days and GFP fluorescence was visualized using ×10 objective Leica DMi8 microscope with Leica LAS X software (Leica). On day 7, the transducted IFP-MSC were sorted based on GFP expression to yield the aCGRP IFP-MSC subpopulation using MoFlo Astrios EQ cell sorter (Beckman Coulter, Brea, CA, USA). After sorting, aCGRP IFP-MSC were seeded on 0.1% gelatin plates (MilliporeSigma, St Louis, MO, USA) and were cultured at 37 °C, 5% (*v*/*v*) CO_2_ until 80% confluency.

### 2.4. Clonogenic Assay of aCGRP IFP-MSC

aCGRP IFP-MSC at passage three (P3) (n = 3) were seeded in 25 cm^2^ flasks in triplicate at a density of 5 × 10^2^ cells per flask in complete medium. Colony-forming unit fibroblasts (CFU-Fs; consisting of at least 50 cells each) were manually enumerated on day 15 after cytochemical staining with 0.01% Crystal Violet (MilliporeSigma, Billerica, MA, USA).

### 2.5. Immunophenotype of aCGRP IFP-MSC

Flow cytometric analysis was performed on P3 aCGRP IFP-MSC (n = 3). Briefly, 2 × 10^5^ cells were labelled with antibodies specific for: CD10, CD73, CD90, CD105 (Biolegend, San Diego, CA, USA), CD146 (Miltenyi Biotec, Auburn, CA, USA), HLA-DR (BD Biosciences, San Jose, CA, USA) and the corresponding isotype controls. All cells were stained with eFluor 780 fixable viability dye (Invitrogen). The fluorescent signal was acquired using a CytoFLEX S (20,000 events) and analyzed with Kaluza analysis software (Beckman Coulter).

### 2.6. Isolating sEVs from aCGRP IFP-MSC

sEVs were isolated from aCGRP IFP-MSC conditioned media by a stepwise ultracentrifugation method. Briefly, conditioned media from aCGRP IFP-MSC cultured in sEVs-depleted Ch-R medium [32] are filtered through a 0.22 µm filter to remove debris and large vesicles, and differentially centrifuged for 2000× *g* for 10 min and ultracentrifuged for 120,000× *g* for 4 h using SW 40 Ti Swinging-Bucket Rotor for Optima XPN ultracentrifuge (Beckman Coulter) [33]. Pre-enriched sEVs preparations were stored in DPBS at −80 °C. Samples from each group were assessed for biophysical and biochemical characterization using flow cytometry and nanoparticle tracking analysis [34].

Pre-enriched sEVs were incubated with the Dynabeads^®^-based Exosomes-Human CD63 Isolation/Detection Reagent (Invitrogen) and purified according to the manufacturer’s instructions for magnetic selection. CD9 (Invitrogen) expression was used to validate sEVs presence in CD63^+^-gated particles by flow cytometry. The specific fluorescent labeling of 20,000 events was analyzed on a CytoFLEX S with Kaluza 2.2.1 analysis software (Beckman Coulter).

Nanoparticle tracking analysis (NTA) (NanoSight NS300, Malvern, Worcestershire, UK) from aCGRP IFP-MSC sEVs was performed for quantity and size determination. All samples were diluted 1:10 in PBS. The following settings were set according to the manufacturer’s software: detection threshold 5; room temperature; number of frames 30 and measurement time 30 s. The size distribution and particle concentration each represent the mean of five individual measurements.

The functional assessment of aCGRP IFP-MSC sEVs by macrophage polarization and cortical neurons neuroinflammation assays was performed in a concentration corresponding to sEVs secreted from 1 × 10^6^ aCGRP IFP-MSC.

### 2.7. miRNA Profile of aCGRP IFP-MSC sEVs

miRNA was extracted from aCGRP IFP-MSC sEVs using Total Exosome RNA and Protein Isolation Kit (Thermo Fisher Scientific) according to manufacturer’s instructions. Total sEVs miRNA (1 μg) was used for first-strand cDNA synthesis with All-in-One miRNA First-Strand cDNA Synthesis Kit (GeneCopoeia, Rockville, MD, USA).

Pre-designed human MSC exosome 166 miRNA qPCR arrays (GeneCopoeia; Appendix A) were performed using 1000 ng cDNA per IFP-MSC sample (n = 2), and processed using StepOne Real-time thermocycler (Applied Biosystems, LLC, Waltham, MA, USA). Data analysis was performed using qPCR results with GeneCopoeia’s online Data Analysis System (http://www.genecopoeia.com/product/qpcr/analyse/ (accessed on 10 January 2024)). Mean values were normalized to small nucleolar RNA, C/D box 48 (SNORD48), expression levels were calculated using the 2^−ΔCt^ method. Putative miRNA interactomes were generated using a miRNet centric network visual analytics platform (https://www.mirnet.ca/ (accessed on 20 January 2024)). The miRNA target gene data were collected from well-annotated database miRTarBase v8.0 and miRNA-gene interactome network refining was performed with 2.0 betweenness cut-off. Values (with 34 cycles cut-off point) were represented in a topology miRNA-gene interactome network using force atlas layout and hypergeometric test algorithm. miRDB online database (http://mirdb.org (accessed on 22 January 2024)) for prediction of functional miRNA targets has been used to correlate highly expressed target genes in macrophages and synoviocytes with specific miRNAs identified by IFP-MSC sEV miRNA profiling. MirTarget prediction scores are in the range of 0–100% probability, and candidate transcripts with scores ≥ 50% are presented as predicted miRNA targets in miRDB [35].

### 2.8. Protein Profile of aCGRP IFP-MSC sEVs

Multiplex protein arrays of 35 cytokines and 41 growth factors (RayBio^®^ C-Series, RayBiotech, Peachtree Corners, GA; Appendix A) were used to determine aCGRP IFP-MSC sEV protein cargos (n = 2). Specifically, 132 μg/mL of aCGRP IFP-MSC sEV protein extract was used for each assay following the manufacturer’s instructions. Data shown represent 30 sec exposure in FluorChem E chemiluminescence imaging system (ProteinSimple, San Jose, CA, USA). Results were generated by quantifying the mean spot pixel density of each array using protein array analyser plugin using ImageJ Fiji software (Fiji/ImageJ, NIH website). Results generated by ImageJ Fiji software were introduced into excel-based analysis software tools for the automatic computation of the extracted numerical data obtained from the array image (https://www.raybiotech.com/tools/array-analysis-tool (accessed on 10 February 2024)). All signal intensities were normalized with the background signal of each array. Also, all signal intensities were normalized to positive and negative signal to reduce batch variability. Finally, the quantified signal intensity for each protein spot is proportional to the relative concentration of the antigen in the sample.

### 2.9. Pathway Analysis

Putative interactomes were generated by the Search Tool for Retrieval of Interacting Genes/Proteins (STRING 11.0; available from: http://string-db.org (accessed on 20 February 2024)) database using interaction data from experiments, databases, neighborhoods in genome, gene fusions, co-occurrence across genomes, co-expression and text-mining. An interaction confidence score of 0.4 was imposed to ensure high interaction probability. K-means clustering algorithm was used to organize proteins into 3 separate clusters per condition tested, discriminated by colors. Functional enrichments related to biological process, Kyoto Encyclopedia of Genes and Genomes (KEGG) pathways, and reactome pathways were presented in radar graphs for proteins tested.

The miRDB online database (http://mirdb.org (accessed on 22 February 2024)) for prediction of functional miRNA targets has been used to correlate highly expressed target genes in macrophages with specific miRNAs identified by aCGRP IFP-MSC sEV miRNA profiling. MirTarget prediction scores are in the range of 0–100% probability, and candidate transcripts with scores ≥ 50% are presented as predicted miRNA targets in miRDB [35].

### 2.10. Culturing HEK 293 with aCGRP IFP-MSC sEVs

HEK293 cells were cultured in growth medium (DMEM/10%FBS/1%PenStrep) and upon 80% confluency were detached, centrifuged and resuspended in 1 mL of growth medium with the addition of aCGRP IFP-MSC sEVs at 30 μg/mL, 50 μg/mL and 80 μg/mL concentrations. Cells/aCGRP IFP-MSC sEVs were incubated at 37 °C, 5% (*v*/*v*) CO_2_ for 1 h and then HEK293 cells were re-cultured for 5–7 days. From this culture, 5 × 10^4^ cells were seeded per well of 12-well plate and cultured overnight. The next day, cells were fixed with 4% paraformaldehyde and incubated in 5% blocking serum for 2 h, followed by overnight incubation with primary antibodies against CGRP (1:500, Abcam) and AAV proteins (1:500, Invitrogen). Finally, cells were washed with 0.1M PBS, incubated in fluorescent secondary antibody (1:250, AlexaFluor, Invitrogen) with DAPI as a nuclear marker, and coverslipped with Vectastain (VectorLabs, Newark, USA). Images were evaluated under 20× objective with confocal microscope (Dragonfly, Andor, South Windsor, USA), digitalized, and analyzed with Image J Fiji software (NIH). To measure and compare the optical density of the CGRP staining between different treatments, at least 3 sections per slide with at least 30 DAPI positive profiles were analyzed. Each section was loaded into Image J Fiji software under CGRP filter, thresholded to the dimmest pixels and the overall integrated density of the section was measured.

### 2.11. Macrophage Polarization Assay

Human monocytes (THP-1, ATCC) were differentiated into macrophages using PMA/IO (Phorbol 12-myristate 13-acetate/Ionomycin) and polarized to M1 macrophages by M1-macrophage generation medium (PromoCell, Heidelberg, Germany). 5 × 10^4^ PMA/IO-stimulated THP-1 (macrophages) were mixed with aCGRP IFP-MSC sEVs (n = 2) per well of 24-well plate and cultured in M1-macrophage generation medium for 2 days. Macrophage polarization status was assessed using a polarization qPCR array (ScienCell, Carlsbad, CA, USA).

RNA extraction from THP-1 cultures was performed using the RNeasy Mini Kit (Qiagen, Frederick, MD, USA) according to manufacturer’s instructions. Total RNA (1 μg) was used for reverse transcription with a SuperScript™ VILO™ cDNA synthesis kit (Invitrogen). A pre-designed 40 gene Human Macrophage Polarization array (GeneQuery™ Human Macrophage Polarization Marker qPCR Array Kit, ScienCell; Appendix A) was performed using 1000 ng cDNA per culture and processed using a StepOne Real-time thermocycler (Applied Biosystems, LLC). Mean values were normalized to GAPDH, expression levels were calculated using the 2^−ΔΔCt^ method. Values were represented in a stacked bar plot for M0, M1, and M2 polarization as the relative fold change of the PMA/IO + THP-1/aCGRP IFP-MSC sEVs to PMA/IO + THP-1 (reference sample, 2^−ΔΔCt^ = X sample/X reference sample).

### 2.12. Cortical Neurons Neuroinflammation Assay

E14 rat cortical neurons were isolated from rat brain according to protocol [36]. 1 × 10^5^ TIC inflammatory/fibrotic cocktail (15 ng/mL TNFα, 10 ng/mL IFNγ, 10 ng/mL CTGF)-stimulated cortical neurons were mixed with aCGRP IFP-MSC sEVs (n = 2) per well of 12-well plate and cultured for 3 days. Cortical neurons neuroinflammation status was assessed using a neuropathic and inflammatory qPCR array (Qiagen).

RNA extraction from TIC-stimulated cortical neurons was performed using the RNeasy Mini Kit (Qiagen) according to manufacturer’s instructions. Total RNA (1 μg) was used for reverse transcription with a SuperScript™ VILO™ cDNA synthesis kit (Invitrogen). A pre-designed 84 gene rat neuropathic and inflammatory array (RT2 Profiler neuropathic & inflammatory array, Qiagen; Appendix A) was performed using 1000 ng cDNA per culture and processed using a StepOne Real-time thermocycler (Applied Biosystems, LLC). Mean values were normalized to ACTB, expression levels were calculated using the 2^−ΔΔCt^ method. Values were represented in a stacked bar plot as the relative fold change of the TIC-stimulated cortical neurons/aCGRP IFP-MSC sEVs to TIC-stimulated cortical neurons alone (reference sample, 2^−ΔΔCt^ = X sample/X reference sample).

### 2.13. Statistical Analysis

Normal distribution of values was assessed by the Kolmogorov–Smirnov normality test. In the presence of a non-normal distribution of the data, one-way or two-way ANOVA were used for multiple comparisons. All tests were performed with GraphPad Prism v7.03 (GraphPad Software, San Diego, CA, USA). Statistical significance was considered *p* < 0.05.

## 3. Results

### 3.1. aCGRP IFP-MSC Generation and Characterization

In this study, we have successfully transduced IFP-MSC cultured in regulatory-complaint medium with GFP-labelled AAV CGRP_8–37_ and expanded them for 7 days in vitro until confluency. Cell sorting of transduced cells resulted in purification of an aCGRP IFP-MSC subpopulation (0.74 ± 0.11%, Figure 2). To build on our previous results [28,29], the present experiments aimed at investigating the effects of AAV CGRP_8–37_ transduction on IFP-MSC basic characteristics. Therefore, following positive selection, aCGRP IFP-MSC were characterized by standard assays, showing similar fibroblast-like morphology, lower clonogenic capacity (72 ± 42 CFU-Fs) and similar immunophenotyping profiles to non-transduced IFP-MSC published in our previous studies [28,29]. Specifically, transduction with AAV CGRP_8–37_ had a significant impact on aCGRP IFP-MSC clonogenicity whereas aCGRP IFP-MSC showed high (>90%) expression for common MSC-defining markers (Figure 2B,C). CD10 expression was similarly high (>95%), whereas CD146 showed reduced expression compared to non-transduced IFP-MSC (24.4 ± 5.6% vs. 75.4 ± 4.5%) [29]. Herein, high CD10 expression levels indicate that aCGRP IFP-MSC show increased functionality in SP degradation. Finally, as expected, HLA-DR expression was absent indicating that transduction with AAV CGRP_8–37_ is not affecting the non-immunogenic profile of IFP-MSC.

aCGRP IFP-MSC sEVs showed high purity with <200 nm sizes, a concentration of 3-4 × 10^9^/mL, and high expression levels of CD9 marker upon ultracentrifugation (111.7 ± 4.3 nm, Figure 2D). Immunostaining analysis of HEK293 cells incubated with aCGRP IFP-MSC sEVs showed a positive signal for AAV proteins and CGRP protein, suggesting successful transfer of AAV CGRP_8–37_ within sEVs (Figure 2E). Immunodensity of CGRP signal was proportional to the total protein content of sEVs used for transfer. The protein concentrations used 30, 50, 80 μg/mL are corresponding to sEVs secreted from 0.3 × 10^6^, 0.5 × 10^6^, and 0.8 × 10^6^ aCGRP IFP-MSC, respectively. Lower values of AAV integrated density but enhanced CGRP concentration were observed at 80 μg/mL total protein of sEVs compared to 50 μg/mL total protein.

### 3.2. aCGRP IFP-MSC sEVs miRNA Cargo Characterization

From 166 MSC-related miRNAs analyzed, 147 miRNA cargos were present in aCGRP IFP-MSC sEVs (Figure 3A). Reactome analysis of the miRNAs present in sEVs showed their involvement in the regulation of mainly six gene groups in target cells related to gene expression, immune system, TGF-β/Wnt/FGFR pathways, cell cycle, NGF and Toll-like receptor pathways, and cellular responses to stress (Figure 3B). Out of the overall 7457 genes in target cells regulated by miRNAs present in sEVs, 497 genes were related to the immune system, 316 genes were related to TGF-β/Wnt/FGFR pathways, and 245 genes were related to NGF and Toll-like receptor pathways. These results indicate the high involvement of aCGRP IFP-MSC sEVs miRNA cargo in the regulation of a plethora of pathways related to immunomodulation and pain signaling.

In aCGRP IFP-MSC sEVs, 19 miRNAs cargos were highly present (hsa-miR-4466, hsa- miR-3665, hsa-miR-6089, hsa-miR-1290, hsa-miR-4516, hsa-miR-7975, hsa-miR-4454, hsa-miR-21-5p, has-miR-4792, has-miR-4488, has-miR-1246, hsa-miR-23a-3p, hsa-miR-100-5p, hsa-miR-4286, hsa-miR-99a-5p, has-let-7b-5p, hsa-miR-107, hsa-miR-574-5p, and hsa-miR-27b-3p) (Figure 4A). From these miRNAs, 7 miRNAs (hsa-miR-6089, has-miR-21-5p, hsa-miR-23a-3p, hsa-miR-99a-5p, hsa-miR-574-5p, hsa-let-7b-5p, hsa-miR-1290) were associated in previous studies with significant anti-inflammatory/immunomodulatory effects in vitro and in vivo. Reactome analysis of 19 miRNAs highly present in aCGRP IFP-MSC sEVs showed their involvement in the regulation of four gene groups in target cells related to the: gene expression, TGF-β/Wnt/FGFR signaling pathways, cell cycle, and NGF signaling (Figure 4A). Therefore, these distinct miRNAs regulate genes involved in the production of pro-inflammatory cytokines, the recruitment of monocytes, and cartilage homeostasis.

The miRDB in silico analysis revealed a functional correlation of identified miRNAs in aCGRP IFP-MSC sEVs with genes involved in M2 macrophage polarization, immunomodulatory and pain signaling, and cartilage homeostasis (Figure 4B). For M2 macrophage polarization the MirTarget prediction score system revealed *MRC1* as a target for hsa-miR-23a-3p (99% probability), *IL10* as a target for hsa-let-7b-5p (76% probability), *VEGFA* as a target for hsa-miR-107 (51% probability), *TGF-β* as a target for hsa-miR-4286 (64% probability), and *CD163* as a target for hsa-miR-4516 (61% probability). For immunomodulatory and pain signaling the prediction scoring algorithm revealed *NGF* as a target for hsa-let-7b-5p (86% probability), *IL12A* as a target for hsa-miR-21-5p (97% probability), *CDH11* as a target for hsa-miR-27b-3p (90% probability), and *TAC1* as a target for hsa-miR-23a-3p (65% probability). Lastly, for cartilage homeostasis the prediction scoring system revealed *WNT* as a target for hsa-miR-4488 (94% probability), *PRG4* as a target for hsa-miR-3665 (73% probability), *FGFR1* as target for hsa-miR-7975 (54% probability), *TGF-β* as a target for hsa-miR-4286 (64% probability), and *EGFR* as a target for both hsa-miR-27b-3p (91% probability) and hsa-miR-1290 (85% probability). Overall, from a clinical standpoint, the potent functionality of these miRNAs supports the notion of developing novel cell-free therapeutics for inflammation/pain reversal based on aCGRP IFP-MSC sEVs.

### 3.3. aCGRP IFP-MSC sEVs Protein Cargo Characterization

We detected multiple immunomodulatory and reparative molecules secreted as a cargo of aCGRP IFP-MSC sEVs (Figure 5A). sEVs showed presence of key immunomodulatory molecules including TIMP-2, IL-8, MCP-1, IL-6, ICAM-1, sTNF-RI, MIP-1β, IL-10, and IP-10. In parallel, key reparative molecules including HGF, VEGF, EGFR, IGFBP-1, βFGF, and IGFBP-6 showed presence in aCGRP IFP-MSC sEVs.

All proteins appeared interconnected at least through one association and all K-means clustering networks demonstrated elevated protein–protein interaction enrichment (*p* < 1 × 10^−16^) and an average local clustering coefficient >0.8 indicating that the proteins used are at least partially biologically connected. In terms of biological processes, various categories were highly affected and presented as % of proteins involved in a category to the total proteins detected. Specifically, regulation of signal transduction (81%), positive regulation of response to stimulus (79%), positive regulation of cell population proliferation (71%), positive regulation of cell migration (55%), positive regulation of protein phosphorylation (57%), regulation of immune system process (38%), and angiogenesis (31%) (Figure 5B). Regarding the type of signaling pathways affected, aCGRP IFP-MSC sEVs have profound effects on PI3K-Akt signaling pathway (59%), MAPK signaling pathway (55%), Ras signaling pathway (52%), Rap1 signaling pathway (43%), cytokine-cytokine receptor interaction (24%), and Jak-STAT signaling pathway (19%) (Figure 5B).

### 3.4. aCGRP IFP-MSC sEVs Effects on Macrophages

PMA/IO-stimulated THP-1 showed similar morphology with and without aCGRP IFP-MSC sEVs treatment (Figure 6A). However, their molecular profiling indicated a strong gene expression shift towards an M0/M2 macrophage polarization (Figure 6A). Most importantly, the expression levels of *CD200R1*, *BMP7*, *IRF4*, *IL10*, *IL12A*, characteristic M2-polarization markers, were strongly induced when macrophages were exposed to aCGRP IFP-MSC sEVs. In parallel, 6 genes related to the M0 phenotype (*HLA-DQA1*, *HLA-DRA*, *HLA-DQB1*, *SOCS3*, *FABP4*, *NFKB1*) were highly upregulated upon aCGRP IFP-MSC sEVs exposure of PMA/IO-stimulated THP-1 cells. In future experimentation, stronger M2 macrophage polarization could be achieved by higher dose of sEVs (Figure 6A).

### 3.5. aCGRP IFP-MSC sEVs Effects on Cortical Neurons

TIC-stimulated cortical neurons showed similar morphology with and without aCGRP IFP-MSC sEVs treatment (Figure 6B). However, their molecular profiling indicated an overall reduced neuroinflammatory profile upon exposure to aCGRP IFP-MSC sEVs (Figure 6B). From 84 genes, only 14 (*SLC6A2*, *IL1B*, *PTGS1*, *EDN1*, *SCN11A*, *CCR2*, *TLR2*, *GDNF*, *CD4*, *OPRD1*, *ACE*, *ALOX5*, *PTGES*, *CCL12*) showed increased expression (>2-fold) compared to TIC-stimulated cortical neurons alone. Interestingly, 4 major genes (*MAPK8*, *CD200*, *MAPK1*, *PTGES3*) involved in neuropathic pain were highly down-regulated (>2-fold) upon exposure to aCGRP IFP-MSC sEVs.

## 4. Discussion

CGRP, a 37 amino-acid peptide, is well documented in playing a key role in the transmission of pain signals in both the periphery and the CNS. Accumulating evidence implicates a role for CGRP in the peripheral mechanisms of OA-mediated pain [1,2,3,4,5]. Multiple knee joint structures are richly innervated by CGRP-expressing sensory neurons [3,37]. A truncated form of the CGRP peptide, CGRP_8–37_, is a selective antagonist that binds to the CGRP receptor with approximately the same affinity as CGRP but does not result in receptor signal transduction.

In this study, we have successfully transduced IFP-MSC which showed high expression of CD10 surface marker. CD10 is a neutral endopeptidase expressed in multiple cell types including MSC [38,39], with enzymatic activity neutralizing various signaling substrates including SP [26,40]. SP is a neuropeptide associated with nociceptive pathways that is secreted by sensory nerve fibers in the synovium and IFP tissues. Upon its secretion it actively affects local inflammatory/immune and fibrotic responses by modulation of cell proliferation, activation and migration to sites of inflammation, and the expression of recruiting chemokines and adhesion molecules [22,23]. Based on this, we previously reported that IFP-MSC acquire a potent immunomodulatory phenotype and actively degrade SP via CD10 in vitro and in vivo [28,29]. Therefore, high CD10 expression levels indicate that aCGRP IFP-MSC show increased functionality in SP degradation.

Similar to our previous studies [30,34,41], aCGRP IFP-MSC sEVs show high purity with <200 nm sizes and high expression levels of CD9 marker upon ultracentrifugation. Importantly, we have shown the successful transfer of AAV CGRP_8–37_ within yielded sEVs. Predominantly, miRNAs present in sEVs are involved in the regulation of TGF-β/Wnt/FGFR pathways [42]. TGF-β is a superfamily of growth factors that bind to activin receptor-like kinases (ALKs) and activate SMAD signaling. In OA, due to elevated TGF-β concentrations to which the chondrocytes are exposed to, the function of TGF-β changes from a factor that blocks chondrocyte hypertrophy (Smad2/3 signaling) to a factor that facilitates chondrocyte hypertrophy (Smad1/5/8 pathway) [43]. Recent evidence has suggested that WNT/β-catenin signaling may also play a role in the pathogenesis of OA. Canonical WNT pathway activation can play a prominent role in the proliferation of osteoblasts leading to osteophyte formation and chondrocytes hypertrophy [44]. FGFR signaling plays an important role in the pathogenesis of OA as specifically FGFR1 is highly expressed in OA chondrocytes. In OA, FGF2 binding to FGFR1 leads to a cascade of events that are typically catabolic in articular chondrocytes, including the activation of mitogen-activated protein kinase (MAPK) pathways [42]. Importantly, miRNAs present in sEVs are also involved in the regulation of NGF and Toll-like receptor signaling. NGF has been discovered to have chemotropic effects on the CNS and have significant role in inflammatory conditions in terms of the sensation of pain and activation of the immune system [45]. In OA, inflammatory stimuli enhance NGF release by chondrocytes whereas NGF is released in a dose and time-dependent relationship to IL-1β [46]. Toll-like Receptors (TLR1-7, TLR9) have also been linked to the pathophysiology of OA as they are upregulated in the synovium of OA joints. Downstream to the upregulation of Toll-like receptors is the increased production of pro-inflammatory molecules such as NF-kB, TNF-α, and IL-1β. This, in turn, recruit’s macrophages and immune cells to the site of the synovium. Toll-like receptor activity is also linked to the initiation of apoptosis; therefore, its high activity in the synovium might be linked to the apoptosis of chondrocytes, progressing the pathophysiology of OA [47,48].

In aCGRP IFP-MSC sEVs, 7 miRNAs cargos highly present were associated in previous studies with significant anti-inflammatory/immunomodulatory effects in vitro and in vivo. Specifically, hsa-miR-6089 targets the TLR4 mediated inflammatory response and inhibits the activation of macrophages through downregulation of IL-6, IL-29, and TNF-α cytokine production [49]. Similarly, hsa-miR-21-5p and hsa-miR-23a-3p are also involved in the regulation of inflammatory cytokine production, such as IL-1β, IL-6 and TNF-α. Deficiency in hsa-miR-21-5p leads to increased recruitment of CD11b+ monocytes/macrophages and an increased inflammatory response [50,51]. Both hsa-miR-99a-5p and miR-574-5p inhibit NLRP3 inflammasome by promoting macrophage autophagy through downregulation of mTOR signaling and by suppressing leukocytes infiltration and downregulating NF-κB [52,53]. Also, miR-107 protects against inflammation and apoptosis of endothelial cells via KRT1-dependent Notch signaling pathway [54] whereas hsa-let-7b-5p regulates neutrophil function and reduces neutrophilic inflammation by suppressing the canonical TLR4/NF-κB pathway [55]. Lastly, hsa-miR-1290 is directly related to the differentiation status of MSC undergoing chondrogenic differentiation indicating its putative chondroprotective and anabolic effects for cartilage regeneration in vivo [56].

We detected multiple immunomodulatory proteins (TIMP-2, IL-8, MCP-1, IL-6, ICAM-1, sTNF-RI, MIP-1β, IL-10, and IP-10) secreted as a cargo of aCGRP IFP-MSC sEVs. The highest present protein, TIMP-2, attenuates the development of inflammation and inflammatory mediated pain via MMP-dependent and receptor-mediated cell signaling mechanisms [57]. Specifically, NF-κB, Nrf2, and CREB pathways are largely involved in TIMP-2-mediated anti-inflammatory effects [58]. When TIMP-2 is knocked out in OA models, there is increased angiogenesis and increased degeneration of the articular cartilage [59]. Also, IL-6 and IL-8 cytokines have been implicated in both pro- and anti-inflammatory actions with previous studies indicating that the IL-6/IL-8 ratio plays a crucial role in the specific polarization of the cellular microenvironment [60]. MCP-1 and ICAM-1 are associated with monocyte recruitment and adhesion in inflamed tissues after pro-inflammatory cytokine activation [61,62]. Importantly, MCP-1 and CCR2 chemokine receptors determine the extent of M2 macrophage polarization by enhancing the production of the anti-inflammatory IL-10 cytokine [62]. Also, studies have illustrated that IP-10 secretion directly correlates with decreased T cell proliferation [63]. Importantly, IL-10 inhibits MHC class II and costimulatory molecule B7-1/B7-2 expression on monocytes and macrophages and limits the production of pro-inflammatory cytokines [64].

Key reparative proteins (HGF, VEGF, EGFR, IGFBP-1, βFGF, and IGFBP-6) showed also presence in aCGRP IFP-MSC sEVs. HGF is a multifunctional growth factor that, like its specific receptor c-Met, is widely expressed in joint tissues. Studies showed that HGF promotes cell proliferation and survival in chondrocytes while inhibiting ECM degradation of articular tissue, both of which are key regulators of degenerative changes in OA [65]. EGFR is critical for the maintenance of homeostasis of the superficial layer of articular cartilage. Animal models lacking EGFR have fewer superficial chondrocytes, less secretion of boundary lubrication, weak mechanical strength at the cartilage surface, and continued OA progression [66]. Interestingly, VEGF secreted by MSC induces endothelial progenitor cell differentiation towards endothelial cells via paracrine actions [67]. IGF-1 promotes cartilage anabolism by inducing proteoglycan synthesis in chondrocytes via PI3K pathway and by stimulation of chondrocyte proliferation via the PI3K and MEK/ERK pathways [68]. The two highly present IGF-1 binding proteins, IGFBP-1 and IGFBP-6, extend the plasma half-life of IGF-1 and therefore enhancing/prolonging its anabolic effects in vivo. The role of βFGF for articular cartilage homeostasis is controversial. However, recent studies have shown that βFGF can induce the expression of TIMP1 gene (which encodes an inhibitor of MMPs) in cartilage and can suppress IL-1-induced expression and activity of ADAMTS4 and ADAMTS5, enzymes that degrade aggrecan in human articular cartilage [69].

In this study, we have shown that aCGRP IFP-MSC sEVs can effectively polarize M1 macrophages to M2 anti-inflammatory phenotype. Most importantly, the expression levels of *CD200R1* [70], *BMP7* [71], *IRF4* [72], *IL10* [73], *IL12A* [74], characteristic M2-polarization markers, were strongly induced upon aCGRP IFP-MSC sEVs exposure. Both *CD200R1* and *BMP7* induce M2 macrophage polarization, whereas *BMP7* significantly increase the expression of anti-inflammatory markers, arginase-1 and IL-10 [71]. *IRF4* has been shown to regulate the expression of glycolytic genes, suggesting a conserved link between mTOR signaling, IRF4 and metabolic reprogramming in immune cells [72]. Also, studies showed that IL-10-induced M2 macrophage phenotype implicate intracellular protein BCL3 [73]. Finally, studies showed that *IL12A* deletion significantly exacerbates macrophage accumulation, polarize macrophages to M1 phenotype and increase the expression levels of macrophage-secreted proinflammatory cytokines [74].

According to preliminary studies, the truncated CGRP peptide, CGRP_8–37_, acts as a CGRP antagonist and can transiently reverse symptoms of migraine and neuropathic pain in animal models [9,10,11]. Preliminary findings in the Sagen et al. lab have shown that direct intraspinal or intrathecal injection of AAV_CGRP_8–37_ can reduce chronic neuropathic pain symptoms in rat spinal cord injury and peripheral nerve injury pain models. In the present study, inflamed cortical neurons exposed to aCGRP IFP-MSC sEVs treatment showed an overall reduced neuroinflammatory profile. Interestingly, 4 major genes (*MAPK8*, *CD200*, *MAPK1*, *PTGES3*) involved in neuropathic pain were highly down-regulated (>2-fold) upon exposure to aCGRP IFP-MSC sEVs. Mitogen-activated protein kinases (MAPKs) are a family of serine/threonine protein kinases that play essential roles in inflammation and tissue remodeling [75,76] and are involved in the modulation of nociceptive signaling and the peripheral/central sensitization produced by intense noxious stimuli [77]. Previous studies have demonstrated that the inhibition of MAPKs produces anti-inflammatory effects in various inflammatory diseases [75]. MAPKs activation contributes to pain sensitization via both neuronal and glial mechanisms that may be mediated by their different isoforms, such as MAPK1 and MAPK8. These two isoforms were downregulated following sEVs treatment. Notably, studies showed that MAPK8 activation in astrocytes contributes to the maintenance of chronic pain [78]. In parallel, MAPK1 has been recognized as key molecule of pain signaling that can induce alone inflammatory pain but cooperates with MAPK3 in sensory neuron survival [79]. In summary, we observed an alteration at the neuroinflammatory molecular profile of cortical neurons upon exposure to aCGRP IFP-MSC sEVs. We acknowledge that the use of cortical neurons to study neuropathic pain is not the optimal cell source. Therefore, based on the present promising data we aim to use dorsal root ganglion neurons, the main knee-innervating nociceptor neurons, to further clarify the anti-inflammatory/analgesic effects of aCGRP IFP-MSC sEVs in vitro. Also, further reduction of the neuroinflammatory profile could be achieved by higher doses of sEVs. Finally, future preclinical studies will determine the effectiveness of aCGRP IFP-MSC sEVs for inflammation and pain reversal in vivo.

Therefore, aCGRP IFP-MSC sEVs strongly indicate their potent immunomodulatory functionality which upon further studies in animal models of OA and other inflammatory models could result in the development of specialized cell-free therapies that overcome regulatory constraints for safe and effective regulation of inflammation and pain in humans.

## 5. Conclusions

Cell sorting of AAV transduced IFP-MSC yielded purified cultures of CD10-bound IFP-MSC expressing GFP tagged gene encoding for a competitive antagonist of CGRP. These cells yielded aCGRP IFP-MSC sEVs with distinct miRNA and protein signatures as cargos. In silico analysis demonstrated that detected miRNAs and proteins regulate multiple pathways and biological processes involved in the control of pain, inflammation, and cartilage homeostasis. Additionally, aCGRP IFP-MSC sEVs effectively polarize M1 macrophages to the M2 phenotype and alter the neuroinflammatory profile of stimulated cortical neurons. Collectively, our data suggest that yielded sEVs can putatively target CGRP signaling involved in inflammation and pain in vivo.

## Figures and Tables

**Figure 1 cells-13-00484-f001:**
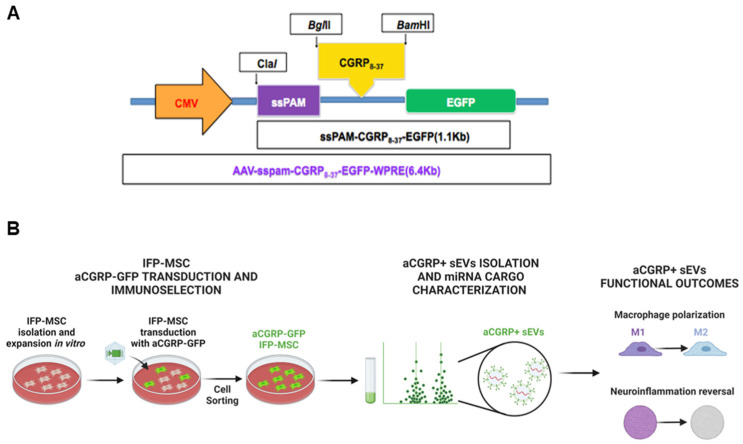
(**A**) Gene construct of adenovirus vector containing the gene for CGRP_8–37_. The CGRP_8–37_ fragment was inserted downstream of the signal sequence of peptidylglycine-amidating monooxygenase (ssPAM/pGEMT). This arrangement facilitates the amidation and secretion of CGRP_8–37_. (**B**) MSC were isolated from human infrapatellar fat pad (IFP), transduced with gene construct, and sorted by FACS cell sorting to generate aCGRP IFP-MSC. aCGRP IFP-MSC sEVs were isolated and characterized for their miRNA cargo. Functional assessment of aCGRP IFP-MSC sEVs was performed by macrophage polarization and cortical neurons neuroinflammation assays.

**Figure 2 cells-13-00484-f002:**
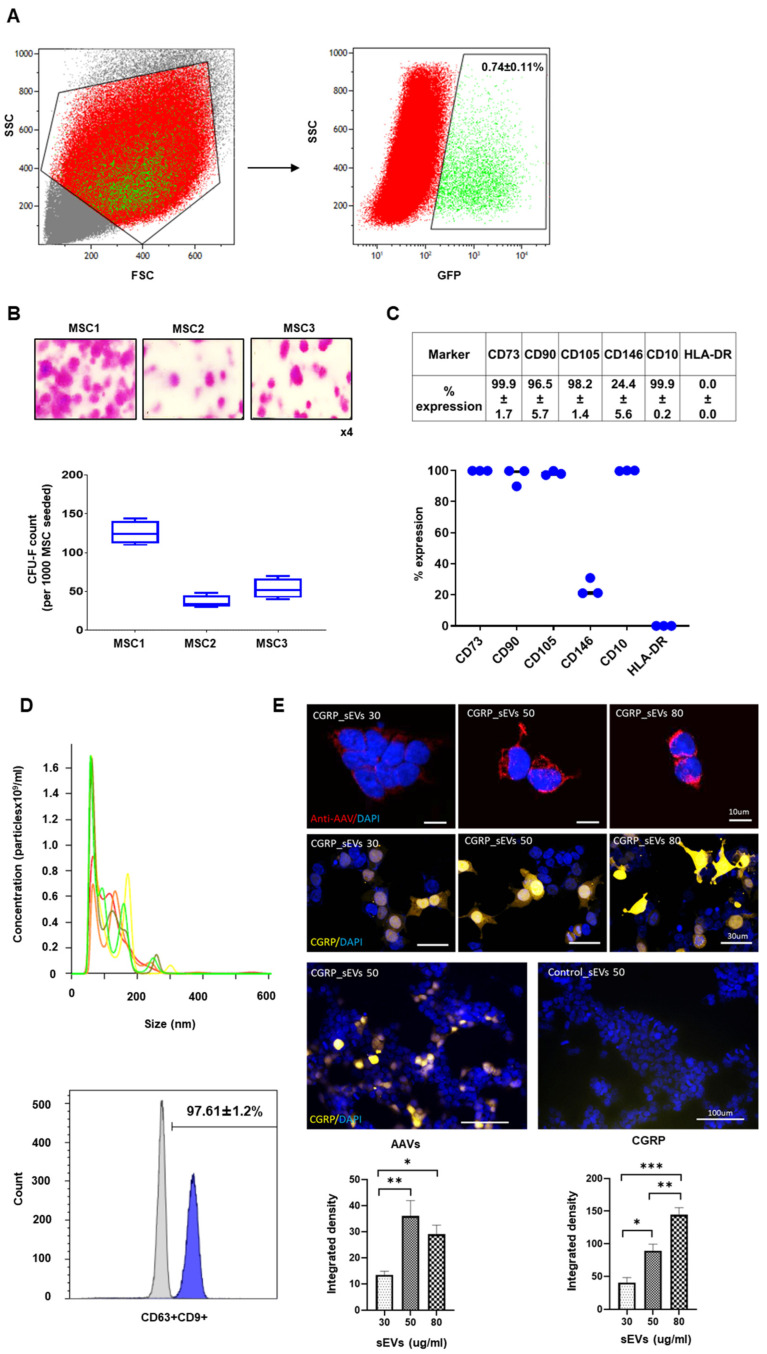
(**A**) FACS sorting of GFP-labelled AAV CGRP_8–37_ transduced IFP- MSC (n = 5). Cell sorting of transduced cells resulted in the purification of an aCGRP IFP-MSC subpopulation. (**B**) aCGRP IFP-MSC showed similar fibroblast-like morphology, but lower clonogenic capacity (72 ± 42 CFU-Fs) to non-transduced IFP-MSC. (**C**) aCGRP IFP-MSC showed high expression levels of common MSC-defining markers (CD73, CD90, CD105, CD146, CD10, HLA-DR). CD10 expression was similarly high (>95%) whereas CD146 showed reduced expression compared to non-transduced IFP-MSC. (**D**) aCGRP IFP-MSC sEVs showed high purity and <200 nm sizes. CD63^+^-selected sEVs showed high positivity for CD9 marker. (**E**) Immunostaining of HEK 293 cells after incubation with aCGRP IFP-MSC sEVs. Top: Detection of AAV protein in HEK cells incubated with different concentration range of sEVs, Middle: Detection of CGRP protein in HEK cells incubated with different concentration range of sEVs, and bottom: Lower magnification of HEK cells after incubation with sEVs from recombinant MSC (left) and control, sEVs from no recombinant MSC. Plots: Top: Integrated density of AAV signal. aCGRP IFP-MSC sEVs were used at 30 μg/mL (30), 50 μg/mL (50) and 80 μg/mL (80) concentrations. * *p* < 0.05, ** *p* < 0.01, and bottom: Integrated density of CGRP signal. * *p* < 0.05, ** *p* < 0.01, *** *p* < 0.001.

**Figure 3 cells-13-00484-f003:**
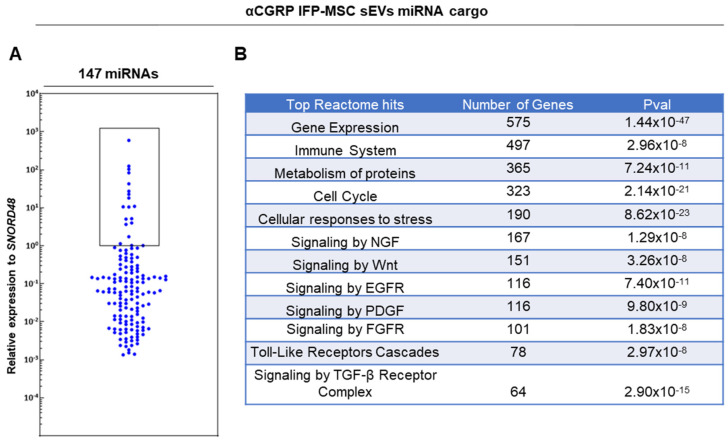
(**A**) 147 distinct miRNA were present in aCGRP IFP-MSC sEVs (n = 2). Nineteen highly present miRNAs are included in black-bordered box. (**B**) miRNAs present in aCGRP IFP-MSC sEVs were involved in the regulation of numerous genes and pathways. Predominantly, miRNAs present in sEVs are involved in the regulation of TGF-β/Wnt/FGFR pathways. Putative miRNA interactomes were generated using a miRNet centric network visual analytics platform. The miRNA target gene data were collected from well-annotated database miRTarBase v8.0 and miRNA-gene interactome network refining was performed with 2.0 betweenness cut-off. Values (with 34 cycles cut-off point) were represented in a topology miRNA-gene interactome network using force atlas layout and hypergeometric test algorithm.

**Figure 4 cells-13-00484-f004:**
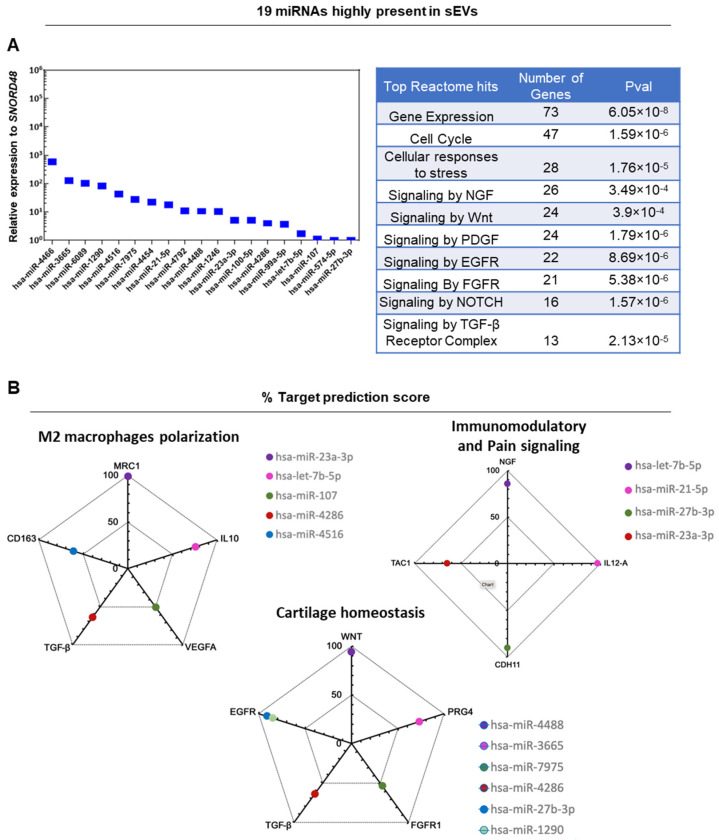
(**A**) In aCGRP IFP-MSC sEVs (n = 2), 19 miRNAs cargos were highly present. These distinct miRNAs regulate genes involved in the production of cytokines, the recruitment of monocytes, and cartilage homeostasis. From these miRNAs, 7 miRNAs were associated in previous studies with significant anti-inflammatory/immunomodulatory effects in vitro and in vivo. Putative miRNA interactomes were generated using a miRNet centric network visual analytics platform. The miRNA target gene data were collected from well-annotated database miRTarBase v8.0 and miRNA-gene interactome network refining was performed with 2.0 betweenness cut-off. Values (with 34 cycles cut-off point) were represented in a topology miRNA-gene interactome network using force atlas layout and hypergeometric test algorithm. (**B**) In silico analysis revealed a functional correlation of identified miRNAs in aCGRP IFP-MSC sEVs with genes involved in M2 macrophage polarization, immunomodulatory and pain signaling, and cartilage homeostasis. The miRDB online database for prediction of functional miRNA targets has been used to correlate highly expressed target genes in macrophages with specific miRNAs identified by aCGRP IFP-MSC sEV miRNA profiling. MirTarget prediction scores are in the range of 0–100% probability, and candidate transcripts with scores ≥ 50% are presented as predicted miRNA targets in miRDB.

**Figure 5 cells-13-00484-f005:**
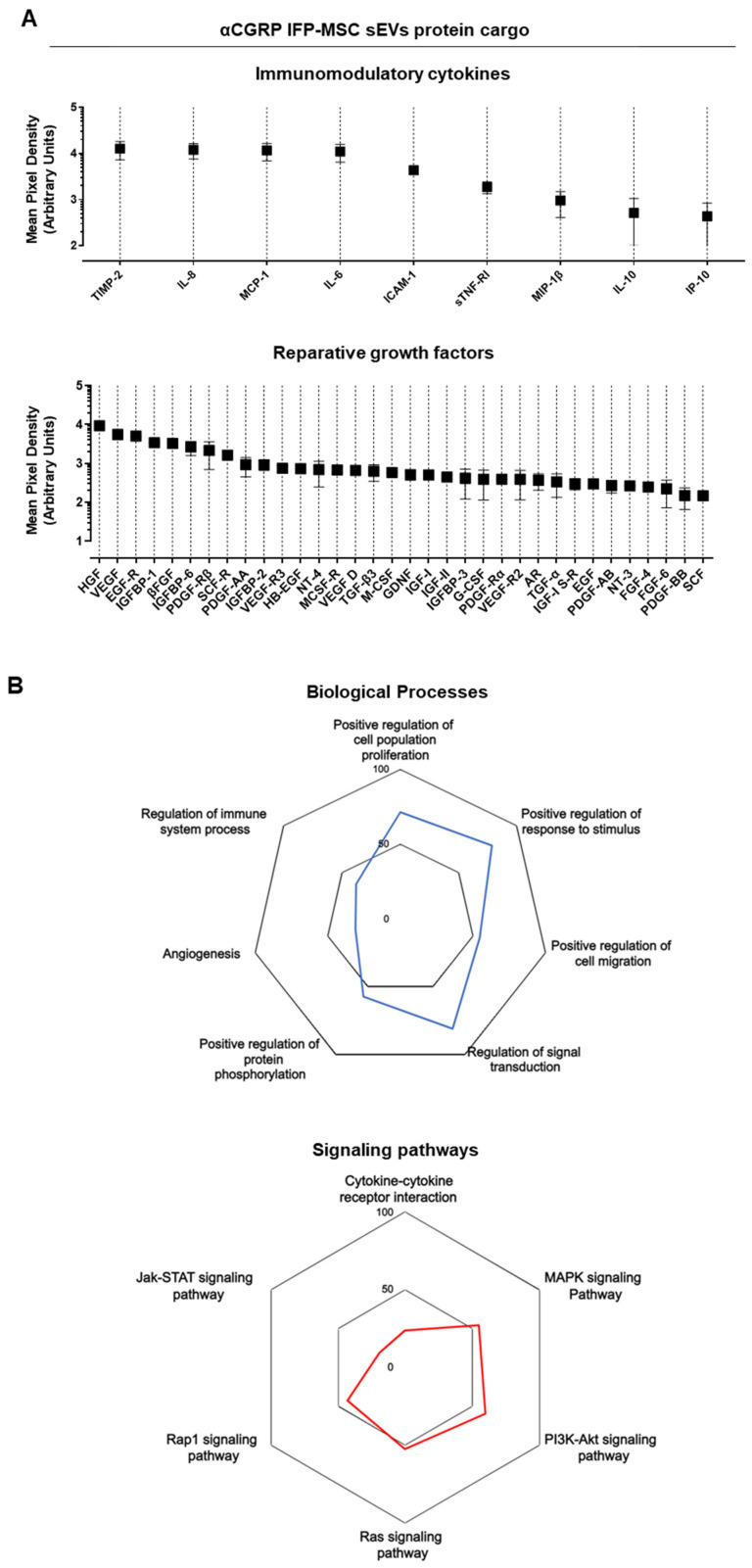
(**A**,**B**) Multiple immunomodulatory and reparative molecules secreted as a cargo of aCGRP IFP-MSC sEVs (n = 2). sEVs showed presence of key immunomodulatory molecules including TIMP-2, IL-8, MCP-1, IL-6, ICAM-1, sTNF-RI, MIP-1β, IL-10, and IP-10. In parallel, key reparative molecules including HGF, VEGF, EGFR, IGFBP-1, βFGF, and IGFBP-6 showed presence in aCGRP IFP-MSC sEVs. These proteins are listed in descending order based on their presence levels within sEVs. The miRDB online database for prediction of functional miRNA targets has been used to correlate highly expressed target genes in macrophages with specific miRNAs identified by aCGRP IFP-MSC sEV miRNA profiling. In terms of biological processes, various categories were highly affected and presented as % of proteins involved in a category to the total proteins detected. aCGRP IFP-MSC sEVs protein cargo have effects on PI3K-Akt signaling pathway (59%), MAPK signaling pathway (55%), Ras signaling pathway (52%), Rap1 signaling pathway (43%), cytokine-cytokine receptor interaction (24%), and Jak-STAT signaling pathway (19%). MirTarget prediction scores are in the range of 0–100% probability, and candidate transcripts with scores ≥ 50% are presented as predicted miRNA targets in miRDB.

**Figure 6 cells-13-00484-f006:**
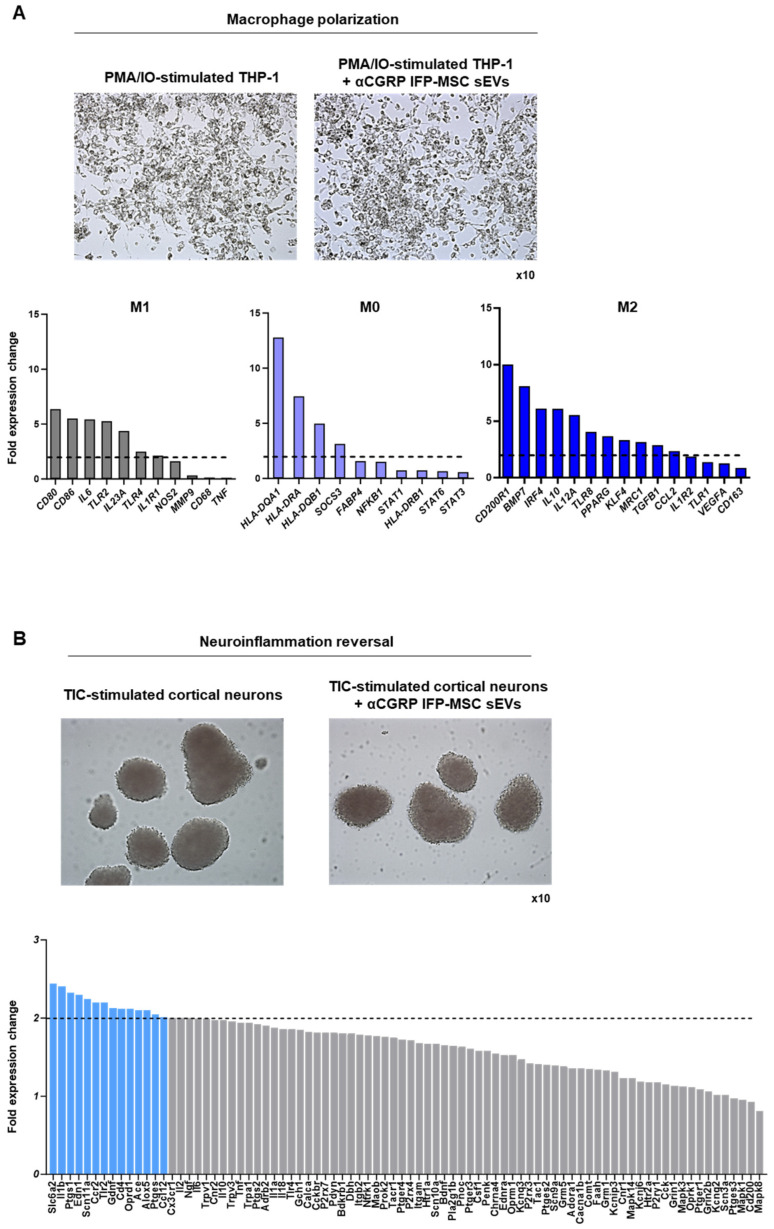
(**A**) PMA/IO-stimulated THP-1 showed similar morphology with and without aCGRP IFP-MSC sEVs treatment (n = 2). Upon exposure to aCGRP IFP-MSC sEVs, PMA/IO-stimulated THP-1 gene expression analysis showed a strong shift towards M0/M2 macrophage polarization. Notably, exposure to aCGRP IFP-MSC sEVs induced significant expression of key M2-polarization markers such as *CD200R1*, *BMP7*, *IRF4*, *IL10*, and *IL12A*. (**B**) TIC-stimulated cortical neurons showed similar morphology with and without aCGRP IFP-MSC sEVs treatment (n = 2). Upon exposure to aCGRP IFP-MSC sEVs, their molecular profiling indicated an overall reduced neuroinflammatory profile upon exposure to aCGRP IFP-MSC sEVs. From 84 genes, only 14 (*SLC6A2*, *IL1B*, *PTGS1*, *EDN1*, *SCN11A*, *CCR2*, *TLR2*, *GDNF*, *CD4*, *OPRD1*, *ACE*, *ALOX5*, *PTGES*, *CCL12*) showed increased expression (>2-fold) compared to TIC-stimulated cortical neurons alone. Blue bars indicate the >2-fold expressed genes. Interestingly, 4 major genes (*MAPK8*, *CD200*, *MAPK1*, *PTGES3*) involved in neuropathic pain were highly down-regulated (>2-fold) upon exposure to aCGRP IFP-MSC sEVs.

## Data Availability

All data needed to evaluate the conclusions in the paper are present in the paper. Additional data related to this paper may be requested from the authors.

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
