# Peer review of "Modification of Mesenchymal Stem/Stromal Cell-Derived Small Extracellular Vesicles by Calcitonin Gene Related Peptide (CGRP) Antagonist: Potential Implications for Inflammation and Pain Reversal"

_cells, 2024, doi:10.3390/cells13060484_

Round 1

Reviewer 1 Report (Previous Reviewer 3)

Comments and Suggestions for Authors

The manuscript is significantly improved and I have no further comments.

Comments on the Quality of English Language

minor edits needed. 

Author Response

We thank the reviewers for the positive feedback. We have now performed minor edits to the manuscript. 

Reviewer 2 Report (Previous Reviewer 1)

Comments and Suggestions for Authors

Dear authors, the quality of the manuscript improved substantially and is now ready for publication.

Author Response

We thank the reviewers for the positive feedback. We have now performed minor edits to the manuscript. 

This manuscript is a resubmission of an earlier submission. The following is a list of the peer review reports and author responses from that submission.

Round 1

Reviewer 1 Report

Comments and Suggestions for Authors

Dear authors,

Thank you for this interesting study which links EV research with potential treatment of pain conditions. However, some major issues were detected which require the author's attention and prevent the acceptance for publication for now. Please find the specific comments below:

Section 2.6: Please describe the ultracentrifugation procedure in more detail to enable other researchers replicating the study. Information such as rotor type (model number, fixed angle or swing bucket) or k-factor are missing. Which buffer was used to store the EVs?

Line 166: The authors state that isolated EVs were biochemically characterised. Although, miRNA profiling and the multiplex protein assay are biochemical methods, those were not able to show the presence of EV protein markers such as CD9, CD63, CD81, Alic, Syntenin or others. This is recommended by the MISEV 2018 guidelines by ISEV. Otherwise, is cannot be claimed that there are EVs present or that EV cargo was analyzed. In addition, it would be interesting to know if aCGRP is actually present in the EV preparation or if it stays with the cells.

Line 178: It should be disclosed which 166 miRNAs were investigated.

Line 198 states “One mL of aCGRP IFP-MSC sEV protein extract was used for each assay” – how much micrograms of protein was used?

Line 200+: The method of obtaining semi-quantitative data via ImageJ should be described in more detail, currently it is described in an unclear way.

Section 2.10: The indicated EV concentrations correspond to how many EVs per cell?

Section 2.11: Why were 20 EVs per cell used for macropage treatment, while only 10 EVs per cell were used to treat neuronal cells?

Line 233: Cells were again fixed after fixation using 4% paraformaldehyde?

Line 265: Why was ACTB used to normalize gene expression, while GAPDH was used for macrophages?

Line 302: What could be an explanation for the reduced CD146 expression?

Line 321: What are the “signs of cytotoxicity”?

Line 322+: “Lower values of AAV integrated density but enhanced CGRP concentration were observed at 80ug/ml total protein of sEVs compared to 50ug/ml total protein.” – Only the shown images were analyzed? The integrated density should be normalized to cell number.

Fig 2B: What does „MSC1“, “MSC2” and “MSC3” mean?

Figure 3A: What is indicated by the black-bordered box?
Figure 3B: What can a reader learn from the presented interactome graphs? The reviewer suggests to remove panel B or replace it with meaningful graphs.

Line 349: The claim “directly affect signaling pathways” is unsubstantiated as no data were shown in this study to support the claim, only a set of identified miRNAs was predicted based on previous experience to be able to modulate signaling pathways. Whether the EV cargo in this study is in a state that allows “directly affect signaling pathways” was not experimentally determined.

Line 373: NF-kB is not a cytokine.

Line 384: Only 16 miRNAs are listed. What does “predominant” mean, how is this defined?

Line 388: The 7 miRNAs should be listed again.

Line 409: Which in silico analysis was performed?

Figure 4B: Similar to figure 3B, the thumb images of the interaction networks do not provide any information for a reader and should be replaced by meaningful graphs.

Line 437: Based on the performed experiments, the authors can only claim that the identified immunomodulatory and “reparative” molecules are associated with EVs, but not that they are within EVs. For this, a protease assay would be required that shows which proteins are protected from proteolytic degradation due to incorporation into EVs. The same applies to the figure legend of Fig 5 on line 492+

Line 479+: What do the given percentages relate to? Eg response to stimulus (79%) – 79% of what?

Line 488: What are or should be the envisioned goals of the “further studies”?

Line 497: How is a “presence level” defined? What is the background/cutoff level? Are the proteins shown all present in relevant amounts or simply “detectable”? “Molecularly” is irrelevant here, has no meaning and should be removed. Again, what do the given percentages relate to?

Line 517/line 548: As the EV doses in this study are defined via microgram protein, it should be stated here how many particles per cell were used (this would represent a true dosing). Microgram protein can be highly misleading, as the extent of co-isolated non-EV proteins is unknown.

Figure 6A: What is the difference between the shown micrographs except that they show stimulated cells with or without treatment?

Figure 6B: The micrographs of the neuronal cells do not resemble differentiated functional neurons, but rather neuroblasts / precursor cells. Why is this an appropriate model to study EV-mediated effects on functional neurons, while there is no evidence that EVs were actually taken up by the cells?

Line 524: What kind of “molecular profiling”? It is stated in the methods section that a “40 gene Human Macrophage Polarization array” was used in the analysis. What were the 40 genes – and how is it possible that 89 genes were investigated by this (line 526)?

Line 529: How much is “highly down-regulated”?

Line 562: “In OA, inflammation, pain, and cartilage degradation are major factors in disease 562 progression.” This is not a conclusion and should be removed.

Line 563: Not AAV transfection, but FACS of AAV transfected IFP-MSCs yielded the claimed cell population.

Line 570: The term EXOs was not used before and should be avoided in favour of “extracellular vesicles”, please have a look in the MISEV 2018 guidelines to find out the reason. Actually, there are no data presented that support the claim: “.. our data suggest that yielded [EXOs] can putatively target both SP and CGRP, two signaling molecules involved in inflammation and pain in vivo, particularly in patients with OA.” – if there are mechanisms modulated that affect SP and CGRP, the effects are probably indirect, as long as the authors can not show direct effects.

Unfortunately, the main hypothesis was not adequately addressed. It was stated that ".. CD10High aCGRP IFP-MSC with CGRP8-37 gene 104 construct can yield sEVs that can target both SP and CGRP activities and provide robust 105 anti-inflammatory and analgesic effects relevant to reversal of OA symptoms". Although, connections between EV-associated miRNA and proteins were made, no direct effects on SP and CGRP were investigated. SP was not even mentioned anymore in the results section. Neither the presence of aCGRP in the EVs was shown, nor the robustness of anti-inflammatory or analgesic effects relevant to reversal of OA symptoms was demonstrated. The reviewer suggests to rescope the aims of the study, as it is nearly impossible to conclude clinical effects based on in silico data and questionable in vitro data.

Reviewer 2 Report

Comments and Suggestions for Authors

Please find my comments below.

1) Abstract exceeds 200 words. The abstract should not have any references included, because it is not a part of the introduction. I recommend to shorten it and edit just to provide the substantial message on the background/introduction, methods, aim, results and conclusion of the study. Please follow the recommendations included in “Instructions for authors” regarding “Cells” journal.

2) line 74 – it was not stated earlier what is NGF. The abbreviation should be expanded.

3) line 95 – sEVs abbreviation should be expanded. Authors included the explanation in the abstract, however the abstract should be considered as an exclusive part of the article summarizing the highlights of each section. As I mentioned, I find the abstract in current form as formulated improperly. Additionally, it is unclear what authors exactly mean by small extracellular vesicles. It is advisable to enrich the Introduction with an explanation what is included into the group of sEVs. What kind vesicles/particles are they? Furthermore, it is not clear what CD10High is.

4) Lines 97-98 – Authors should cite the article referring to the current classification and description of sEVs.

5) Lines 100-103 – The source should be referenced.

6) Line 115 – Does IRB stand for Institutional Review Board? It is recommended to not use abbreviations when unnecessary.

7) Line 139 – It should be specified what Viral Vector Core is.

8) Line 166 – It should be specified what biophysical and biochemical characterization was performed on samples.

9) Line 273 – The p-value does not refer to the level of significance.

10) The section # 3 Require editing. Regular research articles in “Cells” are divided to Introduction, Material and Methods, Results, Discussion and Conclusion. Sections should not be combined. The Discussion fragments should be separated from results and moved into unique section. In current form it is hard to read and understand the true meaning of presented results.

11) Figure 2B – The presented image does not respond to fibroblast-like morphology of cells.

12) Figure 2C – Authors should provide histograms to support these numbers.

13) Figure 2D – The presented plot refers that the sample has 1.6 particles in 1 ml. In words: one point six pieces.

14) In the section 3.1. authors provided disturbingly many self-citations.

15) Authors did not provide any data/methodology which would confirm the origin of isolated sEVs. Samples should be verified with WB and/or TEM.

16) Figure 6A – The upper panel of “A” fragment of figure 6 has no reference in the text of this manuscript. Authors did not mention about presented photos in the main text as well as in the figure caption. The whole “B” fragment of figure 6 has no reference too and it was not discussed either.

 17) The conclusions should not provide any additional, unrelated to conducted experiments information. This section should be devoted to share a substantial summary supported by the results and methodology. No additional story is needed here.

18) Line 570 – Only once in the whole text the Authors have used the wording “EXOs”. In this case this appears like a mistake or laboratory slang. I find this inappropriate.

19) It is not clear why the HEK line was used in the study and what was the used dose of sEVs for treating cells with them and why?

20) It is recommended to check spelling, punctuation and editorial errors.

Reviewer 3 Report

Comments and Suggestions for Authors

The study “Mesenchymal stem/stromal cell-derived small extracellular vesicles for substance P and calcitonin gene related peptide 3 (CGRP) inhibition: Potential implications for inflammation/pain reversal in Osteoarthritis” could potentially be interesting and relevant. However, it is difficult to evaluate in the presented manuscript.

The title is misleading, I couldn’t find the data showing a reduction in Substance P or CGRP expression, there is also no data specifically related to OA, please provide the data or modify the title. The introduction should also be modified accordingly as substance P and CGRP is not specific to OA pain. Alternatively, data showing reduced pain in an OA model should be provided. 

 In Acknowledgment, why is funding described when it didn't support the data generated?

The result and discussion should be separated to allow for a clear and concise description of the new data. In its present form, it is not clear what is new results, results confirming previous data of the group and results published by other teams. Please clearly indicate what has and has not already been published. Already published data could be in supplemental material as proof of that the system is working as expected. 

All figure legends are describing and discussing the data not what was analyzed, how it was analyzed, how many biological replicates that was performed and the statistical significance used. Please correct all figures.  

Why was the 2−ΔCt method and not the 2−ΔΔCt method used? Please normalize the data using the 2−ΔΔCt method and present the data as fold change with the control experiment set as 1.

For dot blot protein arrays; each data point is normalized to positive and negative "spots" to reduce batch variability as described in the manufacturer instructions. Please reanalyze according to the instructions.   

Figure 2, what is 20, 50 and 80 represent? what was analyzed and using what method? In the text, describe what the data is showing. What is A showing, why is the data normalized to SNORD48 and what is the rectangle representing?  How are the Reactome circles with little dots in B related to the table A? Please consider presenting the reactome data in a more comprehensive way.

Figure 3. Its written “From 166 MSC-related miRNAs analyzed, 147 miRNA cargos were present in aCGRP  IFP-MSC sEVs (Figure 3A)”. How was this different to the control cells? Please describe here and in all subsequent figures.

3 B and C; It is written “Out of the overall of 7457 genes regulated by miRNAs present in sEVs”, are the authors describing the genes the miRNA’s that they found in the transfected cells or from EV’s in general? Also, would this also be true for the EVS of non-transfected cells? Please describe, a small up or down-regulation in expression levels might not provide a functional effect.  Were unique miRNAs detected?

Figure 4. it is written “19 miRNAs cargos were predominant” what does this mean, please be more precise and please see my comments in figure 3 and describe how the findings are different to the non-transfected cells.

“In silico analysis revealed a functional correlation of identified miRNAs in aCGRP IFP-MSC sEVs with genes involved in M2 macrophage polarization, immunomodulatory and pain signalling, and cartilage homeostasis (Figure 4C). How is this demonstrated, please describe.  

Figure 5, I can’t understand how the data is presented, it is described as a % difference but to what? Please describe.

Fig 6. Split up the data and present genes involved in M2 polarization and M0 phenotype etc so it is easier to follow the description of the data. Please use the 2−ΔΔCt method so it is easy to appreciate the differences. Please also see my comments in Figure 3 and integrate them here.

What is the difference between the cells in 6 A  +/- Evs and 6B +/- EVs, treated and untreated cells look the same to me. Please clarify.

“Herein, TIC-stimulated cortical neuron molecular profiling indicated an overall reduced neuroinflammatory profile upon exposure to aCGRP IFP-MSC sEVs (Figure 6B). From 89 genes only 14 (SLC6A2, IL1B, PTGS1, EDN1, SCN11A, CCR2, TLR2, GDNF, 526 CD4, OPRD1, ACE, ALOX5, PTGES, CCL12) showed increased expression (>2-fold) compared to TIC-stimulated cortical neurons alone. Interestingly, 4 major genes (MAPK8, CD200, MAPK1, PTGES3) involved in neuropathic pain were highly down-regulated upon exposure to aCGRP IFP-MSC sEVs”.  Will a 2 fold increase result in biologically relevant differences? What is highly down regulated? Please describe the data in exact terms.